# LIME: LEARNING INDUCTIVE BIAS FOR PRIMITIVES OF MATHEMATICAL REASONING

## ABSTRACT

While designing inductive bias in neural architectures has been widely studied, we hypothesize that transformer networks are flexible enough to *learn* inductive bias from suitable generic tasks. Here, we replace architecture engineering by encoding inductive bias in the form of datasets. Inspired by Peirce's view that deduction, induction, and abduction form an irreducible set of reasoning primitives, we design three synthetic tasks that are intended to require the model to have these three abilities. We specifically design these synthetic tasks in a way that they are devoid of mathematical knowledge to ensure that only the fundamental reasoning biases can be learned from these tasks. This defines a new pre-training methodology called "LIME" (Learning Inductive bias for Mathematical rEasoning). Models trained with LIME significantly outperform vanilla transformers on three very different large mathematical reasoning benchmarks. Unlike dominating the computation cost as traditional pre-training approaches, LIME requires only a small fraction of the computation cost of the typical downstream task.

## 1 INTRODUCTION

Inductive bias is essential for successful neural network learning. Many of the breakthroughs in machine learning are accompanied by new neural architectures with better inductive biases, such as locality bias in convolutional neural networks (LeCun et al., 1999), recurrence and memory in LSTMs (Hochreiter and Schmidhuber, 1997), and structural bias in graph neural networks (Scarselli et al., 2008). However, existing designs of inductive biases need to be explicitly encoded in neural architecture. This is sometimes difficult as one may not know the exact mechanism for an abstract ability, in order to describe the architectural bias explicitly. In particular, designing proper inductive bias for abstract concepts such as *mathematical reasoning* becomes an extremely challenging task. Moreover, attempts to design elaborate architectures for reasoning often fall short of the performance of more generic transformer architecture. In this work, we aim to avoid the search for new architectures and investigate whether one can *learn useful inductive bias for mathematical reasoning through pretraining*.

Large-scale unsupervised pretraining of language models revolutionized the field of natural language processing (NLP), improving the state-of-the-art in question answering, name entity recognition, text classification, and other domains, e.g. (Radford et al., 2018; Devlin et al., 2019; Yang et al., 2019; Liu et al., 2019; Raffel et al., 2020; Brown et al., 2020). As a result, pretraining has become a common practice for modern neural network based NLP. One plausible explanation for the benefit of pretraining is that the model can learn world knowledge by memorizing the contents of the natural language corpus. This can be useful in various natural language downstream tasks, such as question answering and text classification. However, there is another potential advantage of pre-training—it may distill inductive biases into the model that are helpful for training on downstream tasks (Brown et al., 2020; Warstadt and Bowman, 2020). We focus on the latter and design pre-training tasks that are intentionally devoid of knowledge and only allow the model to learn inductive bias for reasoning.

Inspired by the logician Charles Peirce (Peirce, 1992), we believe that the following three primitives are the most crucial for reasoning:

1. **Deduction**: the ability to deduce new truths from given facts and inference rules.
2. **Induction**: the ability to induce general inference rules from a set of known facts.
3. **Abduction**: the ability to explain the relationship between the evidences and inference rules.

To endow the models with an inductive bias for mathematical reasoning, we design a synthetic task for each of the three inductive biases. We hypothesize that the transformer networks are flexible enough to learn strong inductive bias from the three synthetic reasoning tasks and consequently improving the downstream tasks. Although such inductive bias may be useful in general reasoning tasks (e.g., NLP tasks), in this work, we focus on mathematical reasoning benchmarks, for which we expect to observe the largest gains. We call training on these tasks LIME – an acronym for "Learning Inductive Bias for Mathematical rEasoning". Note that there is only a limited amount of pretraining data available for formal mathematical benchmarks, therefore the study of generic pre-training techniques is particularly important for the success of machine learning in mathematical reasoning.

We demonstrate that LIME pretrained models provide significant gains across three large mathematical reasoning benchmarks: IsarStep (Li et al., 2020), HOList Skip-tree (Rabe et al., 2020) and MetaMathStep (Polu and Sutskever, 2020). Notably, on the IsarStep benchmark, pre-training improved the top-1 accuracy from 20.4% to 26.9% and top-10 accuracy from 33.1% to 41.0%. Compared to the traditional pre-training tasks, there are two major differences. First, we do not load the input embeddings or the weights in the output layer for finetuning on downstream tasks. This allows us to use the same pre-trained model for a variety of downstream tasks, which can have vastly different vocabularies due to language or tokenization differences. Also, it prevents the transfer of content knowledge from the pretraining to downstream tasks, supporting the evidence of learning inductive biases. Furthermore, pretraining on synthetic tasks require only a fraction of the computational cost of downstream tasks. With only about two hours of training on a single modern GPU, one already obtains all the benefits, in contrast to days of training on a large natural language corpus with hundreds of GPUs/TPUs.

Our method can also be regarded as a form of curriculum learning, in which the model is taught basic, extremely generic but general skills before being trained on the specific problem domain.

To summarize, the contributions of the paper are:

1. Providing the first method to design inductive biases in the form of datasets for mathematical reasoning.

2. Demonstrating significant improvements in the reasoning performance of transformer models on three large mathematical reasoning benchmarks with negligible extra computation cost.

3. By showing how pretraining brings benefits other than learning content knowledge, disentangling the study of its working mechanism.

## 2   Related Work

**Learning Models Applied to Mathematics**    There has been increasing interest in applying deep learning methods to Interactive Theorem Provers (ITP) (Bansal et al.; 2019; Gauthier et al., 2020; Huang et al., 2019; Yang and Deng, 2019; Wu et al., 2020; Li et al., 2020; Polu and Sutskever, 2020). The work that is most related to ours is GPT-$f$ (Polu and Sutskever, 2020). The authors performed pretraining on several natural language corpora and showed significant improvements for an ITP system – MetaMath. Different from ours, they used GPT-style large-scale language modeling pretraining, which dominates the computation cost compared to the downstream task. We, on the other hand, propose pretraining on a few lightweight synthetic tasks costing only a minor fraction of the computation spent on the downstream task.

Lample and Charton (2020) have demonstrated that transformer models can be used for symbolic mathematics by successfully predicting the integrals of formulas from a randomly generated dataset. Similar observations are made for logical problems relevant to verification: that transformer networks can learn the semantics of logics (Hahn et al., 2020). Rabe et al. (2020) have shown that mathematical reasoning can emerge from self-supervised training alone. Li et al. (2020) show that language models can learn to synthesize missing high-level intermediate propositions given a local context. Piotrowski and Urban (2020) used RNNs in automated theorem provers for first-order logic. Wang et al. (2020) explored the use of machine translation to translate between synthetically generated natural language descriptions of proofs and formally represented proofs. Urban and Jakubův (2020) present initial experiments on generating mathematical conjectures with a Transformer model.

Saxton et al. (2019) suggest a dataset for the analysis of mathematical reasoning skills. In contrast to the datasets considered here, their dataset is synthetic, focuses on calculation with concrete numbers, and only contains relatively few symbolic tasks.

**Language Model Pretraining**   The advent of the transformer architecture (Vaswani et al., 2017) and the BERT style pretraining (Devlin et al., 2019) represented a huge improvement in the quality of language modeling. Since then, an explosion of research activity in the area pushed the quality of language models through better pretraining tasks. Where BERT (Devlin et al., 2019) masks out a fraction of the input tokens, later works demonstrated the advantages of masking out subsequences (Song et al., 2019; Dong et al., 2019; Joshi et al., 2020; Raffel et al., 2020; Conneau and Lample, 2019) and whole sentences (Zhang et al., 2020).

Besides the choice of pretraining tasks, the scale of language models is also an important factor. Language models improve in quality and develop new abilities as they grow larger while trained on the same data (Radford et al., 2018; Raffel et al., 2020; Brown et al., 2020).

**Inductive Biases in General**   There have been works studying learning inductive biases in other contexts. In particular, McCoy et al. (2020) studied whether one can learn linguistic inductive biases on synthetic datasets via meta-learning. Papadimitriou and Jurafsky (2020) shows inductive biases learned in music data can be useful for natural language. They further designed several synthetic tasks and showed similar kind of improvements for natural language tasks. From a more theoretical point of view, Xu et al. (2020) formalize an aspect of inductive (architectural) bias under the context of GNNs, with a notation called *architectural alignment*. The architecture is aligned when the architecture can perfectly simulates the ground truth solution. But their work is limited to showing alignment in combinatorial problems, whose ground truth solutions are known. In contrast, our work tries to learn architectural bias by relying on the flexible Transformer architecture and training on synthetic datasets.

**Inductive Biases for Mathematics**   Previous work studying inductive biases for logical reasoning has focused on encoding bias in the neural architecture. Initial works focused on encoding the tree structure of expressions using TreeRNNs (Evans et al., 2018). Graph neural networks are shown to provide a much stronger performance than tree models in premise selection (Wang et al., 2017) and theorem proving (Paliwal et al., 2020). GNNs also scale to larger formulas in SAT (Selsam et al., 2019; Selsam and Bjørner, 2019; Han, 2020), QBF (Lederman et al., 2020), and #SAT (Vaezipoor et al., 2020). Crouse et al. (2019) have shown that pooling mechanisms can have an impact on the performance of GNNs on logical formulas as well. Closely related, Hellendoorn et al. (2020) have shown that it can be helpful to hard-code the tree structure of programs in the attention mask of transformers. Schlag et al. (2019) developed an architecture for encoding relational information using tensor product representation for mathematical reasoning.

## 3   METHODS

In this section, we first discuss the primitives of reasoning, inspired by Peirce's views, and design one synthetic task for each reasoning primitive.

### 3.1   REASONING PRIMITIVES

In Peirce's view, there are exactly three kinds of reasoning: deduction, abduction, and induction. Deduction is known as the workhorse for mathematics. It is the process of deriving new facts by applying logical inference rules to known facts or premises. On the other hand, abduction and induction can be thought of as the inverses of deduction. If we call the premise used in deduction as *Case*, its logical rule as *Rule*, and its conclusion as *Result*, then abduction is equivalently the inference of a Case from a Rule and a Result, while induction may be said to be the inference of a Rule from a Case and a Result. We summarize the three reasoning primitives in the following table:

| Reasoning Primitives | Inference Map |
|---|---|
| Deduction | Rule, Case $\rightarrow$ Result |
| Abduction | Rule, Result $\rightarrow$ Case |
| Induction | Case, Result $\rightarrow$ Rule |

To give an example, we let Rule be "All the beans in this bag are white", Case be "These beans are from this bag", and Result be "These beans are white". Deduction is to derive the fact that these beans are white (Re) from knowing all the beans from this bag are white (R) and these beans are from this bag (C). Abduction explains why the beans are white (Re) from knowing that all the beans in the bag are white (R) – because these beans must be from the bag (C). Lastly, induction aims to provide a general principle to observing the fact that the beans are white (Re) and they come from this bag (C), which is that all the beans in the bag must be white (R). We refer to Peirce (1992) and Bellucci and Pietarinen (2015) for more elaborate discussions on the primitives of reasoning.

Mathematical reasoning exhibits nontrivial uses of these reasoning primitives. Deduction happens when one needs to derive new valid statements from the given premise (Case) and theorems in the library (Rule). Abduction is used to postulate conjectures from the known facts and theorems, allowing one to decompose the challenging theorem into subgoals for proof. Induction, the ability to extract general principles from known facts and theorems is also one of the major activities of mathematical reasoning. It is used when one derives theorems from special cases and proposes new definitions and general frameworks to encapsulate existing knowledge.

## 3.2 LIME SYNTHETIC TASKS FOR REASONING PRIMITIVES

We design three synthetic tasks inspired by the three reasoning primitives. As discussed in the previous section, all of the reasoning primitives consist of three essential elements: Rule, Case, and Result. Inspired by this, we first design a method to generate those elements. Once they are generated, we can construct tasks that predict one element from the other two. In the following, we describe one simple way to generate those three elements, though we acknowledge that there are many other possible approaches.

We require two types of symbols: 1. *math symbols*, 2. *rule symbols*. In general, these symbols can take any forms (e.g., integer representations). But for the ease of discussion, we will think of math symbols as the union of those operators used in mathematics (e.g., "$+ - * = ()\&$") and lower case letters (e.g., $a, b, c \dots$), and rule symbols as upper case letters (e.g., $A, B, C \dots$). We now construct Rule, Case, and Result in order:

1. **Rule** is a randomly sampled string that consists of i) rule symbols and ii) math symbols. The length of the string is randomly sampled from a range. For instance, a randomly sampled rule can be: $A * A + B = C$ with rule symbols $A$, $B$, and $C$.

2. **Case** is a dictionary that represents substitutions. For each rule symbol used in the Rule string, we sample a random string of random length that consists of math symbols. This forms a dictionary, whose keys are all rule symbols, and the values are the corresponding sampled string. To illustrate, following the previous example, for each $A$, $B$ and $C$, we sample a random string to form a dictionary as: $\{A : a, \ B : b, \ C : d + e\}$.

3. **Result** is the outcome of the substitution. For each rule symbol in the Rule string, we replace it with the corresponding value stored in the Case dictionary. This gives rise to the Result string. As per the previous example, we now substitute $A$ with $a$, $B$ with $b$, and $C$ with $d + e$ into the Rule string, generating the Result string: $a * a + b = d + e$.

After Rule, Case, and Result are generated, we can construct three tasks for deduction, abduction, and induction respectively. We define the three synthetic tasks as follows:

- `Deduct`: **Source:** Rule string and Case dictionary. **Target:** Result string.

- `Abduct`: **Source:** Rule string and Result string. **Target:** Case dictionary.

- `Induct`: **Source:** Case dictionary and Result string. **Target:** Rule string.

We also consider a task called `Mix`, which is a uniform mix of three tasks. Namely, during generation, we randomly select a task and sample an example from that task. To formulate them as sequence to sequence tasks, we represent the Case dictionary also as a string, e.g., "$\{A : a, \ B : b, \ C : d + e\}$". An example of `Abduct` using the examples of Rule, Case, and Result above is to predict the target $\{A : a, \ B : b, \ C : d + e\}$ from the source $A * A + B = C$  $a * a + b = d + e$.

Pre-training on our synthetic tasks can be seen as a form of skip-component learning. There are three essential components: Rule, Case and Result, and we skip one of them and use the remaining

two elements to reconstruct the missing one. Past work has shown that learning to predict missing words (Devlin et al., 2019), subsequences (Song et al., 2019; Raffel et al., 2020), or subtrees (Rabe et al., 2020) are strong pre-training tasks.

### 3.3 SYMBOL-AGNOSTIC REPRESENTATION

In order to solve the synthetic tasks, the model needs to distinguish which set of symbols can be substituted (rule symbols). As a result, the model may memorize information about the symbols that is irrelevant to the inductive biases encoded in the task. To prevent such memorization, we propose a way to make the synthetic tasks agnostic to the choice of symbols.

We first note that the choice of symbols is irrelevant to our synthetic tasks. To avoid symbol-specific memorization, for each training and evaluation example, we randomly sample two sets of symbols to be used in Rules and in the rest of the example. But for the `Abduct` task, the model needs to know which symbols are replaced by the Rule part of the example and which symbols are in the Result language. We simply list the split of the symbols used in the example at the beginning of the input string, marked by two special symbols, `<Rule>` and `<Math>`. They are followed by the original source string. The target string remains unchanged. For example, the previous example in the `Abduct` task becomes,

Source: `<Rule>` $A\ B\ C$ `<Math>` $* + = a\ b\ d\ e$ `` $A * A + B = C$ `` $a * a + b = d + e$

Target: $\{A : a,\ B : b,\ C : d + e\}$

In our implementation, we use integers to represent symbols. Specifically, for each example, we sample two disjoint sets of integers from the set $\{1, \ldots, S\}$ to represent the math symbols and the rule symbols, where $S$ is the size of the vocabulary. In our experiments, we sample 44 math symbols and 24 rule symbols for each problem. The complete pseudo-code of generating the symbols, Rule, Case, and Result for one task example is provided in Appendix Algorithm 1.

## 4 EXPERIMENTS

In this section, we present results on three large mathematical reasoning tasks that are especially useful in the context of automated theorem proving. Our results show significant gains in learning inductive biases from synthetic tasks. We have selected three tasks to cover three different styles of interactive theorem provers: The HOL-Light (skip-tree) corpus was created from very high-level tactic-based proofs, but it is less interpretable than IsarStep's declarative style corpus. We also evaluate the next proof-step prediction task on the `set.mm` library of MetaMath, which consists of very granular, basic proof steps. Namely, the proof steps are more predicable and average proof lengths have significantly increased.

### 4.1 EXPERIMENT DETAILS

**LIME Pretraining**  We generate datasets of our synthetic tasks for pretraining: `Deduct`, `Abduct`, `Induct`, `Mix`. For pretraining of IsarStep, we used a vocabulary size $S$ of 1000. For the other two downstream tasks, we used a vocabulary size of 100. The reason we used different vocabulary sizes was that we found (cf. appendix) the discrepancy in vocabulary size affects the performance of a downstream task if it has a very large vocabulary size (IsarStep has 28K). We use 44 math symbols and 24 rule symbols. The length of the Rule string is sampled from 5 to 20, the length of the string for each substitution (the values of Case dictionary) is sampled from 2 to 8. We used word-level tokenization for all the tasks. We pretrained the model for 20K updates. For tasks with larger vocabulary size (i.e., 1000), we found the learning became more difficult. Hence we used a curriculum learning scheme: we first trained the model for 10K steps on the same task with a vocabulary size of 100, then continue training for another 10K step on vocabulary size of 1000. The pretraining was done on a single Nvidia Tesla T4 GPU with 4 CPU cores for 2 hours. We set the maximum number of tokens in a batch to 4096, and accumulate four batches of gradients for one parameter update. We used the Adam optimizer (Kingma and Ba, 2015) with learning rate $3 \cdot 10^{-4}$. We used a dropout rate of 0.1 and label smoothing (Szegedy et al., 2016) with a coefficient 0.1.

Table 1: Test top-1, top-10 (%) accuracy on the IsarStep task.

| Model | Top-1 Acc. | Top-10 Acc. |
|---|---|---|
| No pretrain (Li et al., 2020) | 20.4 | 33.1 |
| HAT (Li et al., 2020) | 22.8 | 35.2 |
| LIME `Deduct` | 24.7 | 37.7 |
| LIME `Abduct` | 26.7 | **41.0** |
| LIME `Induct` | 23.9 | 38.8 |
| LIME `Mix` | **26.9** | 40.4 |

Table 2: Test top-8 Accuracy on Skip-Tree HOList (%).

| Model | Equation completion | Hard type inference | Missing assumptions | Easy type inference |
|---|---|---|---|---|
| No pretrain (Rabe et al., 2020) | 46.3 | 95.0 | 41.8 | 95.9 |
| LIME `Deduct` | 50.3 | 94.8 | **47.9** | 97.0 |
| LIME `Abduct` | 48.4 | 94.8 | 46.1 | 96.3 |
| LIME `Induct` | 44.8 | 94.9 | 42.6 | 96.4 |
| LIME `Mix` | **51.7** | **95.6** | 46.1 | **97.6** |

**Fine-tuning** For all the downstream tasks in this section, when loading the pretrained models for fine-tuning, we do not load in the vocabulary embeddings nor the output layer weights. For the downstream task IsarStep and MetaMathStep, we used four Nvidia Tesla T4 GPU with 16 CPU cores for training. We set the maximum number of tokens in a batch to 4096, and accumulated four batches of gradients for one parameter update. We trained the model for 200K updates. We used the Adam optimizer, and we searched over the learning rates $\{3 \cdot 10^{-4}, 7 \cdot 10^{-4}\}$, and warmup steps $\{4000, 8000\}$. We used a dropout rate of 0.1 and label smoothing with a coefficient 0.1. For the HOList skip-tree task, we used TPUs for running the experiments. We used a batch size of 256 sequences and trained the model for 1 million updates.

**Architecture** All experiments used the transformer base model from Vaswani et al. (2017), i.e. 512 hidden size, 2048 filter size, 8 attention heads. For the IsarStep and MetaMathStep task, we used 6 layers for both the encoder and decoder, implemented using fairseq (Ott et al., 2019). For the HOList skip-tree experiment, we used a somewhat modified transformer architecture with 8 encoder and 4 decoder layers of the same size as above in which the self-attention and attention over the encoder output were merged.

**Evaluation** During training, we kept track of the best validation tokenized BLEU score [1], and we used the model with validation BLEU for evaluation on the test set. We report top-1 and top-10 accuracies. We consider an output sequence as correct if it matches the target sequence exactly. We performed a beam search with width 10. The top-1 accuracy is then defined as the percentage of the best output sequences that are correct. The top-$n$ accuracy is defined as the percentage of target sequences appearing in the top $n$ generated sequences.

### 4.2 ISARSTEP

The IsarStep task is taken from Li et al. (2020). IsarStep is a task of predicting the missing intermediate propositions given surrounding propositions to bridge the gap between the goal and the current state of the proof. The dataset was mined from the public repository of formal proofs of the Isabelle proof assistant (Paulson, 1994). Unlike HOList and MetaMath, IsarStep contains mostly declarative proofs, a proof style close to humans' prose proofs. The dataset has a broad coverage of undergraduate and research-level mathematics and computer science theorems. There are 820K, 5000, 5000 sequence pairs for the training, validation, and test sets with a maximum of 800 tokens in source sequences and 200 tokens in the target sequences. Following Li et al. (2020), during training, we use 512 as the

---

[1] https://github.com/pytorch/fairseq/blob/master/fairseq/tasks/translation.py#L396

Table 3: Test top-1, top-10 (%) accuracy on the MetaMathStep task.

| Model | Top-1 Acc. | Top-10 Acc. |
|---|---|---|
| No pretrain | 67.7 | 76.5 |
| LIME `Deduct` | 68.8 | 77.4 |
| LIME `Abduct` | 68.8 | 76.1 |
| LIME `Induct` | **69.9** | **78.0** |
| LIME `Mix` | 69.1 | 77.9 |

maximum length for both the source and target, and truncated those that exceed the length to 512. For reporting, we evaluate all 5000 test examples regardless of their lengths.

The results on the IsarStep task for four pretrained models and the baseline transformer model without pretraining is shown in Table 1. We also include another baseline, HAT transformer introduced in Li et al. (2020), which is a specially designed hierarchical transformer architecture tailored to this task. We see the pretrained model achieved substantial improvement over the model trained from scratch as well as HAT. Notably, the model that was pretrained on `Abduct` improved the top-10 accuracy from 33.1% to 41.0%, for almost 8% absolute improvement. The model pretrained on `Mix` performed the best on top-1 accuracy, improving the baseline by 6.5% accuracy. We also showed the validation BLEU scores along training in Figure 1. We can see that the pretrained models learned much faster than

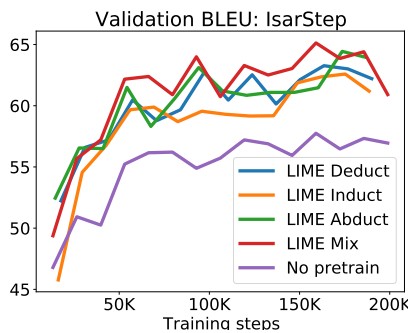

Figure 1: Validation BLEU along training on the IsarStep task.

the model trained from scratch. With around 50K steps of updates, the pretrained model already obtained better BLEU scores than the best score achieved by the un-pretrained model. Moreover, since the downstream task requires 200K steps of training with 4 GPUs, the amount of computation spent on pretraining is only 2.5% of the downstream task, strongly demonstrating the efficiency of the proposed pretraining method.

### 4.3 HOLIST SKIP-TREE

As the second mathematical reasoning benchmark we consider the HOList skip-tree evaluation tasks by Rabe et al. (2020). These tasks include two variants of type inference, predicting under which assumptions theorems hold, and completing equalities. All source expressions for these tasks are taken from the validation set of the theorem database of the HOList proof logs (Bansal et al.). The evaluations are done on a random sample of 1000 instances from the full evaluation sets. We initialized the model parameters with the pretrained weights and then repeated the experiments by Rabe et al. (2020). That is, we trained the models for up to 1M parameter updates on the training set with batch size 256 and repeat the evaluation every 100K steps. In Table 2 we present the best result from these 10 evaluation runs. We see a significant improvement in these reasoning tasks when the models are initialized with the pretrained weights. Notably, on equation completion and missing assumptions task, we improved the beam search (with width 8) exact match rate performance from 46.3% to 51.7% and 41.8% to 47.9%. Note that this is despite the amount of pretraining compute cost being negligible: it takes less than 1 percent of the cost of the downstream task training. Pretraining used 1/20 number of the update steps (50K vs 1M) with 8 (and 4) times smaller batches (pretraining has much shorter sequence lengths, 128 vs. 1024 and 512, respectively).

### 4.4 METAMATHSTEP

Compared to other ITPs, MetaMath is a low-level proving system: each proof step makes only a small step towards the goal. As such, each proof contains many more proof steps than in other ITPs: with 37,000 theorems in the human-written theorem library, there are around 3 million proof steps.

Table 4: Comparisons to other pretraining tasks on IsarStep task.

| Model | Top-1 Acc. | Top-10 Acc |
|---|---|---|
| No pretrain (Li et al., 2020) | 20.4 | 33.1 |
| LIME `Mix` | 26.9 | 40.4 |
| Pretrain on MetaMathStep | 23.1 | 35.7 |
| Pretrain on WMT En-De | 17.2 | 30.3 |

We extract the proof steps and use them to construct a sequence-to-sequence task following Polu and Sutskever (2020) (their proof step training objective).

In this task, the model is asked to generate PROOFSTEPS given a GOAL, namely, the GOAL string is the source input, and PROOFSTEPS is the target output. We follow Polu and Sutskever (2020) and use their string representation for the GOAL and the PROOFSTEPS. Instead of using subword tokenization in Polu and Sutskever (2020), we use a character-level representation for our task. Following Polu and Sutskever (2020), we split theorems into train/valid/test theorems of size 35K, 1K, 1K, and associate all proof steps of a theorem with that split. For each dataset, we filter examples with lengths longer than 1024. This reduced the total number of proof steps to 1.4 million. For validation and test set, we randomly sample 3000 examples out of 40K (after filtering) and perform validation and test evaluations on them. In Table 3 we present the impact of pretraining on our synthetic reasoning tasks on MetaMathStep. We also observe gains from pretraining on this dataset, with the model trained on `Induct` task achieving 2.2% top-1 and 1.5% top-10 test accuracy improvement. Similarly, as for the IsarStep task, the computation spent on pretraining is only 2.5% of the downstream task.

## 5  Ablation Studies

In this section, we perform ablation studies. Additional ablation studies can be found in Appendix C.

### 5.1  Pretraining on Formal Reasoning and Natural Language Tasks

Here we investigate how LIME compares to pretraining on natural language or existing formal reasoning datasets. In this set of experiments, we pretrained three models on `Mix`, MetaMathStep, and on the WMT 2016 English-to-Germany (WMT En-De) translation task, and then we fine-tuned and evaluated these models on the IsarStep task. We pretrained the model on MetaMathStep and WMT EN-DE for 200K steps with 4 GPUs, which is 40 times more computation spent than on LIME. Due to the mismatch between vocabularies of the pretraining task and the downstream task, we do not load the vocabulary embeddings nor output layer weights. The results in Table 4 show that pretraining on MetaMathStep did provide gains, though significantly smaller than gains provided by LIME `Mix`, despite their 40 times higher computational cost. Moreover, pre-training on WMT translation had even a negative effect on the performance. We also conducted an analogous experiment with an evaluation on the MetaMathStep, which we present in Appendix C.

### 5.2  Do we need vocabulary embeddings for fine-tuning?

As mentioned earlier, we did not load in the vocabulary embeddings from the pretrained models when we switched to fine-tuning on downstream tasks. Even without loading the vocab embeddings, the pretrained models still improved the performance. In this ablation study, we investigate how much this decision has affected the results and whether vocabulary embeddings can help improve the performance even further. We performed the comparisons on IsarStep. The task contains a token vocabulary of size 28336. We generated new synthetic tasks for the same vocabulary size, such that we can load the vocabulary embeddings and output layers when initializing the model for IsarStep. Table 5 shows that this led to similar performance. This aligns with our expectation that the model should not learn content specific knowledge that is potentially stored in the vocabulary. These weights turn out to be non-essential for the final performance, supporting the evidence that the transformer learns inductive biases from the pretraining task.

Table 5: Whether one needs to load vocabulary embeddings and output layer weights on IsarStep tasks.

| Model | Top-1 Acc. | Top-10 Acc |
|---|---|---|
| No pretrain (Li et al., 2020) | 20.4 | 33.1 |
| LIME `Mix` | 26.9 | 40.4 |
| LIME `Mix` + Loading All Weights | 26.7 | 40.6 |

## 6    DOES LIME ENCODE INDUCTION, DEDUCTION AND ABDUCTION?

Although LIME has shown to achieve substantial improvements across various benchmarks, it is not entirely clear that the specific synthetic tasks necessarily enforce the reasoning ability of induction, deduction and abduction. We would like to note that deduction, induction, and abduction are high-level and philosophical concepts, and serve only as an inspiration for us to design the synthetic tasks. We do not expect the model will necessarily learn exactly these three capabilities. After all, we have chosen a particular implementation of "Case", "Rule" and "Result". Furthermore, we also design tasks mimic proof steps in formal theorem proving (see the rewrite task in  Appendix B.1), which also achieved excellent results. Nevertheless, we believe LIME is a first step towards building reasoning inductive biases, and provides many inspirations and directions for future work.

## 7    CONCLUSION

In this work, we encoded inductive biases for mathematical reasoning in the form of datasets. We created three synthetic tasks inspired by three reasoning primitives of deduction, induction, and abduction. We demonstrated that pretraining on these tasks (LIME) significantly improved the performances across three mathematical reasoning benchmarks. Notably, LIME requires negligible computation compared to the downstream task, unlike being the dominating factor in previous pretraining methods. Our work naturally poses many future research questions. Could the primitive tasks provide similar gains for NLP tasks? Are there similar primitive tasks for natural language reasoning? We also look forward to disentangling the effects of pretraining between learning content knowledge and inductive bias for all downstream tasks to better understand pre-training.

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

## APPENDIX A    SYNTHETIC TASK GENERATION PSEUDOCODE

---

**Algorithm 1**

---

1: **function** GENERATE_TUPLE( Vocabulary size $S$)
2:     Vocabulary $\mathcal{V} \leftarrow \{1, 2, \ldots, S\}$.                               ▷ Use an integer representation of symbols.
3:     Math symbol set $\mathcal{M} \leftarrow$ SAMPLE($\mathcal{V}$, $n$=44, replacement=False).          ▷ Sample 44 distinct symbols.
4:     Rule symbol set $\mathcal{R} \leftarrow$ SAMPLE($\mathcal{V} \backslash \mathcal{M}$, $n$=20, replacement=False).         ▷ Sample 20 distinct symbols.
5:     Rule $R \leftarrow$ SAMPLE($\mathcal{M} \bigcup \mathcal{R}$, $n$=RANDOM(5,20), replacement=False).          ▷ Sample a sequence of symbols of length between 5 and 20.
6:     Case dictionary $C \leftarrow \{\}$.
7:     **for** $s$ in $\mathcal{R}$ **do**
8:         Case dictionary $C[s] \leftarrow$ SAMPLE($\mathcal{M}$, $n$=RANDOM(2,8), replacement=True).   ▷ Sample a sequence of symbols for each rule symbol, of length of length between 2 and 8.
9:     **end for**
10:     Result $R' \leftarrow$ Rule $R$.                               ▷ Set result string $R'$ to be the same as rule string $R$.
11:     **for** $s$ in $\mathcal{R}$ **do**
12:         SUBSTITUTE($R'$, $s$, $C[s]$).        ▷ Substitute every rule symbol $s$ in result string $R'$ with previously randomly sampled string $C[s]$.
13:     **end for**
14:     **return** Math symbol set $\mathcal{M}$, Rule symbol set $\mathcal{R}$, Rule $R$, Case $C$, Result $R'$.
15: **end function**

---

## APPENDIX B    OTHER SYNTHETIC TASKS

In this section, we give descriptions of other variants of the synthetic tasks we considered than the ones introduced in the main paper.

### APPENDIX B.1    REWRITE AND REWRITE_MULTISTEP

We propose a rewrite task, inspired by the rewrite tactic used in interactive theorem provers. The `Rewrite` task requires the model to rewrite a string according to a rule transformation. One example of the task is:

Source: $a + b - c$ `` $A + B = B + A$

Target: $b + a - c$

"$A + B = B + A$" is the rule transformation, which is applied to the LHS string "$a + b - c$". The model needs to predict the RHS string as the result of the rule application, i.e., $b + a - c$. Besides rule symbols and math symbols, we also require the third set of symbols, named as "string symbols". For the ease of our discussion, we we will think of math symbols as the union of those operators used in mathematics (e.g., "$+ - * = ()\&$"), rule symbols as upper case letters (e.g., $A, B, C \ldots$), and string symbols as lower case letters (e.g., $a, b, c \ldots$). We first sample a random string as the LHS string, consisting of math symbols and string symbols (e.g., $a + b - c$). We sample a sub-string of the LHS string, and replace the string symbols in the sub-string with rule symbols. For example, we sample and obtain the substring $a + b$ from $a + b - c$, and we replace $a, b$ with rule symbols $A, B$. This then forms the LHS of the rule transformation, $A + B$, with the substitution dictionary $\{A : a, B : b\}$. We then sample the RHS of the rule transformation from the union of rule symbols $A$ and $B$, and all math symbols, e.g., $B + A$. This gives the rule transformation $A + B = B + A$. We substitute the value of the substitution dictionary for each rule symbol in the RHS rule, and then substitute back to the original LHS string to obtain $b + a - c$. The task example is constructed by using the LHS string and the rule transformation as the source input, and use the result of the rule transformation as the target.

We further introduce a multi-step version of the rewrite task: `Rewrite_multistep`. In this task, the source may contain more than one rewrite rule, and the target is the result of applying all the rewrite rules in a sequence. This task is motivated from the need to perform multi-step planning in

Table 6: Test top-1, top-10 (%) accuracy on the IsarStep task.

| Model | Top-1 Acc. | Top-10 Acc. |
|---|---|---|
| No pretrain (Li et al., 2020) | 20.4 | 33.1 |
| HAT (Li et al., 2020) | 22.8 | 35.2 |
| LIME `Deduct` | 24.7 | 37.7 |
| LIME `Abduct` | 26.7 | 41.0 |
| LIME `Induct` | 23.9 | 38.8 |
| LIME `Mix` | 26.9 | 40.4 |
| LIME `Rewrite` | 26.0 | 38.6 |
| LIME `Rewrite_multistep` | **28.6** | **43.9** |
| LIME `Induct_v2` | 25.6 | 39.8 |
| LIME `Induct_v3` | 25.0 | 38.8 |
| LIME `Induct_rewrite` | 25.8 | 39.5 |

mathematical reasoning tasks. During pre-training, for each training example, we uniformly sample the number of rewrite steps from 1 to 5.

### APPENDIX B.2   OTHER VARIANTS OF `INDUCT` TASK

We introduce three other variants of the `Induct` task.

1. `Induct_v2`: We move the Case dictionary from the source input to the target output. This makes the task significantly harder, which requires the agent to synthesize a rule and a possible explanation (Case) to explain the Result.

2. `Induct_v3`: Instead of providing the Case dictionary, we provide two Result strings, coming from the same Rule. Namely, we sample two Case dictionaries, and applying each to the Rule string to obtain two Result strings. Both Result strings are used as source, and the target is the Rule string.

3. `Induct_rewrite`: We also create a "induction" version of the `Rewrite` task. In this task, the source is the LHS string concatenated with the RHS string, that is the result of the rewrite. The target is the rewrite rule that is used to do the rewrite.

### APPENDIX B.3   A FULL COMPARISON OF ALL SYNTHETIC TASKS

In this section we present a full comparison for all synthetic tasks. We followed the training protocol in 4.1 and evaluate the method on IsarStep. The results are reported in Table 6. We can see that the `Rewrite_multistep` achieved the best performance across all synthetic tasks, surpassing the baseline by 8.2% for Top-1 accuracy and 10.8% for Top-10 accuracy. This indicates the inductive bias for long horizon reasoning encoded in `Rewrite_multistep` is very useful for the reasoning task.

## APPENDIX C   MORE ABLATION STUDIES

### APPENDIX C.1   DOES THE VOCABULARY SIZE MATTER?

In this section, we investigate whether the vocabulary size $S$ in the synthetic task generation algorithm has an effect on the performance. We used the REWRITE task for the experiment in this section. We generated datasets of various vocabulary sizes, 100, 512, 1000, 5000, 25000. We used the same curriculum learning for pre-training as described in 4.1 on larger vocabulary sizes: first training on the `Rewrite` task of vocabulary size 100 for 10K steps, then training on each individual dataset for another 10K steps. We compare the performance on the downstream task Isarstep. The results are presented in Table 7. We see that when the vocabulary size is equal or larger than 512, the performance were similar. The smallest vocabulary size 100 obtained the worst performance among all, and all the other four models achieved similar BLEU scores. The model trained on the largest vocabulary achieved best performance on top-1 accuracy and top-10 accuracy. The results show there is a non-trivial effect of the vocabulary size of the synthetic task to the performance of the

downstream task. Hence we use vocabulary size of 1000 for all the experiments in the main paper. We leave investigations of the causes to future work.

Table 7: Vocabulary sizes' effects on the IsarStep task.

| Model | Top-1 Acc. | Top-10 Acc |
|---|---|---|
| No pretrain | 20.4 | 33.1 |
| LIME on `Rewrite`, $S = 100$ | 24.1 | 37.5 |
| LIME on `Rewrite`, $S = 512$ | 25.4 | 38.8 |
| LIME on `Rewrite`, $S = 1000$ | 26.0 | 38.6 |
| LIME on `Rewrite`, $S = 5000$ | 25.8 | 38.5 |
| LIME on `Rewrite`, $S = 25000$ | 27.4 | 40.9 |

## APPENDIX C.2    PRE-TRAINING ON ISARSTEP FOR METAMATHSTEP

Following Section 5.1, we performed pre-training on IsarStep for MetaMathStep. The result is shown in Table 8. In contrast to MetaMath helping IsarStep, we see that pretraining on IsarStep task did not help the downstream task MetaMathStep. We hypothesize that this could be due to MetaMathStep task is closer to the LIME tasks than IsarStep, and hence providing more gains than the opposite direction. We leave investigations to the future versions.

Table 8: Pretraining on IsarStep for the MetaMathStep task.

| Model | Top-1 Acc. | Top-10 Acc. |
|---|---|---|
| No pretrain | 67.7 | 76.5 |
| LIME `Mix` | 69.1 | 77.9 |
| Pretrain on IsarStep | 67.0 | 76.1 |

## APPENDIX C.3    DOES LIME HELP LSTMS?

In this section, we investigate if LIME also helps other architectures than transformers. In particular, we applied LIME to two LSTM based architectures: 1. vanilla LSTM, 2. LSTM with attention mechanism. The vanilla LSTM is a stacking LSTM with 4 layers, each with 1000 cells, and 1000-dimensional embeddings. The LSTM with attention architecture is taken from Luong et al. (2015), also with 4 layers, 1000 cells and 1000-dimensional embeddings. We evaluate on the IsarStep task, and compared a model trained from scratch and a model pre-trained on LIME `abduct` task. We used the same training protocol as described in 4.1. The results are shown in Table 9, along with the results on transformer. We observe that LIME improved LSTM as well as LSTM with attention, but the improvements were small compared to transformer. Specifically, if we compare Top-1 accuracy, we can see that LIME improved LSTM from 5.5% to 6.9%, LSTM with attention from 12.3% to 13.4%, and transformer from 20.4% to 26.7%. This observation is aligned with our hypothesis that the transformer is a malleable architecture and hence it is capable of learning architectural inductive biases from datasets. This is mainly attributed to the potential of learning dynamic attention graphs in self-attention layers. We note that this still warrants further investigation as the performance of these architectures are not at the same level, and that may also lead to different improvements.

Table 9: Comparing LIME's benefits on LSTMs on the IsarStep Task

| Model | Top-1 Acc. | Top-10 Acc. |
|---|---|---|
| LSTM | 5.5 | 11.3 |
| LSTM + LIME `Abduct` | 6.9 | 14.3 |
| LSTM + attention | 12.3 | 22.7 |
| LSTM + attention + LIME `Abduct` | 13.4 | 26.3 |
| Transformer | 20.4 | 33.1 |
| Transformer + LIME `Abduct` | 26.7 | 41.0 |

