# OpenReview forum: "LIME: Learning Inductive Bias for Primitives of Mathematical Reasoning"
_ICLR.cc/2021/Conference — Reject_

### Official Review · AnonReviewer1 · 2020-10-26
**Review: LIME**

**Rating:** 6
**Confidence:** 2

**Review:**

With the aim of learning inductive biases for deep neural net architectures, this paper presents three synthetic experiments for learning primitive forms of mathematical reasoning in theorem provers. The overall idea is inspired by Pierce’s view that these primitives are deduction, abduction, and induction. The synthetic tasks are built upon a simple arithmetic language with a source and a target. Using those tasks for pretraining a transformer model, the authors show the merit of this methodology in several mathematical reasoning experiments.

I am not an expert in transformer models and pre-training techniques, so it is possible that I did not understand some parts in Section 4 & 5. Overall, I think that the idea of training a learner with primitive forms of inference is interesting for improving its performance in mathematical reasoning. The experimental results corroborate the relevance of this approach.

My main comment lies in the specification of synthetic tasks (Section 3.2). Here, the authors are using an ad-hoc arithmetic language upon which deduction, induction, and abduction tasks are defined. But this ad-hoc language has no formal semantics, so we cannot formally capture the primitive forms of reasoning.

Contrastingly, in Section 3.1, the three reasoning primitives identified by Pierce are well-defined because they are captured by the semantics of first-order logic. For example, take the abductive mode of reasoning:

Rule: $\forall x, Bag(x) \rightarrow White(x)$ (All the beans from this bag are white)

Result: $White(o)$ (These beans are white, where $o$ is an object in the Herbrand universe)

Case: $Bag(o)$ (These beans are from this bag)

From the rule and the result, we can indeed infer by abduction that the case is an explanation of the result since:

(i) $Bag(o) \wedge \forall x, Bag(x) \rightarrow White(x) \not\models \bot$ (consistency), and

(ii) $Bag(o) \wedge \forall x, Bag(x) \rightarrow White(x) \models White(o)$ (consequence).

But in Section 3.2, the arithmetic language is given without any semantics, and hence, we cannot define a clear, unambiguous form of logical consequence ($\models$). Therefore, the notion of abduction is here very unclear.

To sum up, the synthetic tasks proposed by the authors might indeed help in learning an inductive bias capable of improving theorem provers, but there is a discrepancy between the logical notions of deduction, abduction, and induction defined by Pierce (and more generally in the Knowledge Representation literature), and the reasoning primitives (essentially some forms of pattern matching) presented in the synthetic tasks.

---

> ### Author Response · Authors · 2020-11-20
> **Thank you for your inputs! Induction, deduction and abduction are philosophical concepts, only served as inspiration for our designs.**
>
> We thank the reviewer for your efforts. The following are our answers to specific questions:
>
> "But this ad-hoc language has no formal semantics."
>
> We agree this language does not have formal semantics. However, we note that deduction, induction, and abduction are high-level and philosophical concepts, and serve only as an inspiration for us to design the synthetic tasks. As we remarked at the beginning of section 3.2, “we design three synthetic tasks inspired by the three reasoning primitives”. We do not expect the model will necessarily learn exactly these three capabilities. We also design tasks whose format mimics proving steps in theorem proving (see the rewrite task in Appendix B.1 for one example), which achieve very good results.
>
> In addition, we intentionally did not constrain our synthetic tasks to be grounded in any specific formal logic. We think any formal logic would make the agent prone to overfitting to the particular task distribution. In our synthetic tasks, we do not limit the form of Rule, and Case, allowing the agent to generalize to a broader range of task representations, and hence focusing on learning the essential inductive biases needed to solve such tasks.

---

> > ### Author Response · Authors · 2020-11-23
> > **Does our rebuttal address your concern?**
> >
> > We wonder if our rebuttal have addressed your concerns. If not, we are happy to clarify further. Please let us know. Thank you very much.

---

> > > ### Comment · AnonReviewer3 · 2020-11-23
> > > **Comment from different reviewer**
> > >
> > > Hi AnonReviewer1, I (AnonReviewer3) had the same concern that the synthetic tasks cannot be definitely shown to encode the intended reasoning primitives. After reading the author response, I believe that they make a reasonable argument, namely that:
> > > (i) These tasks are not directly instantiating these primitives, but are instead inspired by these primitives.
> > > (ii) The performance gains make it clear that these tasks have imparted some useful inductive bias; we cannot be certain that this bias relates to the intended 3 primitives, but it's a plausible hypothesis that the bias does relate to those primitives.
> > >
> > > While I believe that this argument is reasonable, I think it would be good for the authors to make sure that the paper overall reflects this argument, rather than the stronger claim that these tasks do instantiate the three primitives. As the authors point out, the start of Section 3.2 says "inspired by", which is in line with this argument; but I believe that other places make a stronger claim - e.g., the abstract says that these tasks "require the model to have these three abilities", whereas that probably should say "are intended to require the model to have these three abilities." Moreover, given that both of us had this concern, it might be worth it to add a paragraph explicitly addressing this concern. E.g., the heading of the paragraph could be "Do these tasks require induction, abduction, and deduction?", and then the body could describe the concern and the authors' response to it.

---

> > > > ### Author Response · Authors · 2020-11-23
> > > > **Paper has been updated to address the common concern following your suggestions.**
> > > >
> > > > Hi AnonReviewer3 and AnonReviewer1,
> > > > Thank you for your great suggestions! We have updated the paper accordingly: 1. change the wording in the abstract, and conclusion. 2. Added a section (section 6) to discuss this:
> > > >
> > > > "DOES LIME ENCODE INDUCTION, DEDUCTION AND ABDUCTION?
> > > >
> > > > Although LIME has shown to achieve substantial improvements across various benchmarks, it is not
> > > > entirely clear that the specific synthetic tasks necessarily enforce the reasoning ability of induction,
> > > > deduction and abduction. We would like to note that deduction, induction, and abduction are high-level and philosophical concepts, and serve only as an inspiration for us to design the synthetic tasks.
> > > > We do not expect the model will necessarily learn exactly these three capabilities. After all, we have
> > > > chosen a particular implementation of "Case", "Rule" and "Result". Furthermore, we also design
> > > > tasks mimic proof steps in formal theorem proving (see the rewrite task in Appendix B.1), which also
> > > > achieved excellent results. Nevertheless, we believe LIME is a first step towards building reasoning
> > > > inductive biases, and provides many inspirations and directions for future work."
> > > >
> > > > We hope this addresses your concern. If not, please let us know.
> > > > Thank you very much again!

---

> > > > > ### Comment · AnonReviewer3 · 2020-11-23
> > > > > **Thank you!**
> > > > >
> > > > > Thank you, the added section does address this concern for me!

---

> > > > > ### Comment · AnonReviewer1 · 2020-11-23
> > > > > **Re: Paper has been updated to address the common concern following your suggestions.**
> > > > >
> > > > > Yes, indeed, thanks for your effort. With this additional paragraph in hand, we understand that LIME is paving the way towards more refined reasoning primitives for inductive biases.

---

### Official Review · AnonReviewer3 · 2020-10-28
**Interesting, creative, and well-motivated approach for giving mathematical inductive biases to a model.**

**Rating:** 8
**Confidence:** 4

**Review:**

In this work, the authors introduce a method called LIME for imparting certain mathematical inductive biases into a model. The structure of the approach is to first pretrain the model on synthetic tasks that are designed around three principles of mathematical reasoning: deduction, induction, and abduction. Each of these pretraining tasks is a sequence-to-sequence mapping involving 3 basic components of reasoning (Rule, Case, and Result), where two of these three components are provided as input and the third component is the target output. After pretraining on these tasks, the authors fine-tune on 3 different proof datasets, and find that the pretraining almost always improves performance, sometimes by a large margin.

Strengths:
1. This approach is creative and thought-provoking; pretraining is an important topic in ML nowadays, and this paper gives several interesting insights about how to structure pretraining. Therefore, publishing this paper at ICLR could help inspire others to use and develop improved variations of pretraining.

2. One aspect of the pretraining that I found particularly impressive was how the authors found such clear improvements from such small amounts of pretraining. This is in stark contrast to the usually massive pretraining datasets that are used, and stands as an especially strong piece of evidence for the model’s usefulness.

3. The experimental setup is well-motivated, drawing on a principled analysis of the problem domain.

4. The paper is overall clearly written and clearly structured.

5. There were some interesting discussion points and ablation studies analyzing the approach in more detail. I particularly liked the discussion about how loading the vocabulary weights had little effect, showing that the inductive biases that were imparted were abstract in nature. It was also useful to see that LIME was more useful than other pretraining tasks, ruling out the possibility that you could get similar improvements from just any pretraining task.

Weaknesses:

1. Part of the paper’s motivation for imparting inductive bias through a dataset, rather than through an architecture, is that designing an architecture “strongly requires human insight.” This is true, but LIME also seems to strongly rely on human insight, so this point is not a benefit for LIME over architectural approaches. This is not a huge problem, but it does not seem like a great motivation for LIME.

2. Related to the previous point, it would be good to discuss the fact that the usefulness of LIME may be limited by the need to design the right pretraining task(s). As Table 4 shows, the nature of the pretraining task is very important; and although the authors were able to create some successful pretraining tasks for mathematical reasoning, it might be harder to create similarly useful tasks for larger-scale tasks in, e.g., language or vision. Again, this is not a huge problem, but I think it at least deserves some discussion.

3. Though the goal of the approach (if I am understanding correctly) is to give inductive biases for induction, deduction, and abduction, the paper gives no direct evidence that it has done so: The authors create an approach *intended* to impart certain inductive biases, and this approach improves performance on 3 tasks that plausibly would benefit from those biases. But this result does not necessarily mean that the model has the inductive biases that were intended to be imparted; it’s possible that LIME imparted some other inductive biases that are also useful for mathematical reasoning but that are not related to induction, deduction, and abduction. Thus, there is a bit of a gap between the motivation and the actual experiments.

4. It’s not entirely clear to me that the specific tasks (Deduct, Induct, Abduct) will necessarily enforce the types of reasoning that they are intended to enforce. For instance, consider the following input/output example: {A : a, B: b, C: d+e} <s> A A + B = C -> a a + b = d + e. Such an example is intended to show deduction, but it could instead be viewed as induction (where A A + B = C is the Result, a a + b = d + e is the Rule, and the Case dictionary should be read in reverse, treating the values as keys and the keys as values). Thus, related to the previous point, I think there is some concern that the LIME tasks may not necessarily encode the intended primitives. The results show that the LIME tasks clearly encode something useful, but it’s not clear exactly what useful things they encode.

Recommended citations: (you definitely don’t need to include all of these or even any of these, but I’m pointing to them just in case they’re useful):

1. You already cite the GPT-3 paper (Brown et al.), But it might make sense to cite it in a second place as well, for the sentence where you say “However, there is another potential advantage of pre-training--it may distill inductive biases into the model that are helpful for training on downstream tasks.” Another paper you can cite for this point is this one: Can neural networks acquire a structural bias from raw linguistic data? https://arxiv.org/pdf/2007.06761.pdf

2. Like your approach, the following paper also uses carefully-constructed synthetic datasets as a way to impart targeted inductive biases into a model. (However, they use these tasks for meta-training, not pre-training): Universal LInguistic Inductive Biases via Meta-Learning. https://arxiv.org/pdf/2006.16324.pdf. This paper might also be useful as an example of how you can address the last two points I listed under weaknesses, as this paper gives examples of how to test whether a model has some specific inductive biases; the paper I linked to in the previous bullet (Warstadt and Bowman) also does this. (However, adding such analyses might be more work than would be doable for a camera-ready).

3. It might be good to cite Peirce when first mentioned in the intro; right now, the citation to Peirce is buried deep in the paper, after he has already been discussed at length.

4. Some more potentially-relevant examples of architecturally encoding inductive biases for math: https://arxiv.org/pdf/1910.02339.pdf, https://arxiv.org/pdf/1910.06611.pdf

Other comments (these are not things that have affected my assessment. Instead, they are just comments that I think might be helpful in revising):

1. Note that there is another approach in ML called LIME, which could potentially cause confusion. It’s completely up to you, but I would consider renaming to avoid confusion. Here is the other LIME by Ribeiro, Singh, and Guestrin: https://dl.acm.org/doi/pdf/10.1145/2939672.2939778?casa_token=VrGSeKoqOnkAAAAA:tmzXq2uCWkUVyPdd9ytCNK4LSdRfIwsIeX4hd8EMkjnjevZ4d-rCeIIM7acIRWGtQlQemUqDlAJx-Q

2. Abstract: “neural architecture” should be “neural architectures”

3. Abstract: “on three large very different mathematical reasoning benchmarks” should be “on three very different large mathematical reasoning benchmarks”

4. Abstract: I did not understand what “dominating the computation” meant until I read the rest of the paper.
The intro says “It is commonly believed that the benefit of pre-training is that the model can learn world knowledge by memorizing the contents of the natural language corpus.” This statement seems strong - I am more inclined to think that much of the benefit comes from learning linguistic structure, not world knowledge. So it might be safer to reword as saying “One plausible explanation for the benefit of pretraining is…”

5. Page 3 says “the BERT pretraining objective,” which suggests that BERT is the objective. But BERT is a model, not an objective; the objectives are masked language modeling and next-sentence prediction.

6. Table 1: The formatting of the table makes it look like the first two rows are numbers copied from Li et al. But from the prose of your paper, and from looking at Li et al, I’m pretty sure that these numbers are from your own re-implementation. Is that correct? If so, it might be best to format the table different - using the citation within the body of the table gives a strong suggestion that the numbers come from Li et al., in my opinion.

7. Table 4 and Table 5: In the caption, say what task these results are for, so that the table can be understood on its own.

8. Please double check the references: Several of them seem to only list authors, title, and year when there is at least an arXiv version that could be listed as well. E.g., “Mathematical reasoning via self-supervised skip-tree training”, “Enhancing sat solvers with glue variable predictions”, “transformers generalize to the semantics of logics”. Also, where possible, cite a paper’s actual publication venue instead of arXiv - e.g., the Raffel et al. T5 paper appeared in JMLR, not just arXiv.

Summary: Overall, I am rating this an 8 because I find the strengths compelling but think that the weaknesses in framing hold the paper back from an even higher score. I would consider increasing the score if those weaknesses were addressed, though those weaknesses are deep enough that it would be hard to properly address them in time.

---

> ### Author Response · Authors · 2020-11-20
> **Thank you for your insightful, deep questions, and many suggestions to improve the text. Our responses are as follows. Part I.**
>
> We thank the reviewer for your efforts, great questions, and many editorial suggestions. We are glad that the reviewer finds the paper interesting, and the work is meaningful. In the following are our answers to specific questions:
>
> “Part of the paper’s motivation for imparting inductive bias through a dataset, rather than through an architecture, is that designing an architecture “strongly requires human insight” "
>
> We thank the reviewer for pointing this out. We find the original sentence in the paper does not convey what we had in mind. We agree both approaches require human insight. Our point is that sometimes it is harder to specify explicitly what the shape of the ideal architecture is (architectural engineering) than to specify what we want the architecture to be capable of (LIME). Mathematical reasoning is such an example. It is unclear what the optimal architecture should look like for doing induction, deduction, and abduction. But it is more straightforward to specify tasks that require these capabilities. However, we totally believe that we still may require further architectural advances beyond current architectures. Our paper is intended to show there is an alternative to architecture engineering for designing inductive biases, and one should embrace it whenever convenient. We have modified the text accordingly in the updated version, and hope this addresses the reviewer’s concern.
>
>
> “It would be good to discuss the fact that the usefulness of LIME may be limited by the need to design the right pretraining task(s).”
>
> We believe that logical/mathematical reasoning is a very general task. While LIME may not directly improve, say, image recognition or natural language processing, it is a generic improvement of reasoning skills, and thereby it is clearly on a path that may have a huge impact across many different fields of science and technology.
> In the short term, we believe that the conceptual contributions presented in this paper may directly affect other fields within machine learning: the design of synthetic, lightweight pretraining tasks to induce useful biases into transformers may turn out to be replicable in other fields.

---

> > ### Author Response · Authors · 2020-11-20
> > **Thank you for your insightful, deep questions, and many suggestions to improve the text. Our responses are as follows. Part II.**
> >
> > “Though the goal of the approach (if I am understanding correctly) is to give inductive biases for induction, deduction, and abduction, the paper gives no direct evidence that it has done so.”
> >
> > We would like to thank the reviewer for posing this meaningful and important question. We want to point out two indirect pieces of evidence.
> >  - Firstly, the transformer achieved almost 99% accuracy on the induction, deduction, and abduction tasks. This shows that the pretrained model is capable of doing these three synthetic tasks.
> >  - Secondly, we would like to go back to the original hypothesis that the transformer is a malleable structure that allows the learning of inductive biases in the datasets. If the hypothesis is true, then learning these three tasks does endow the model with its corresponding architectural inductive biases. As we showed in Appendix C.3, LIME improved transformers much more than LSTM or LSTM with attention, and this provides evidence for our hypothesis.
> >
> > Lastly, we would like to remark that this is a difficult question to answer, as we do not know what the architecture would be to have such inductive biases. This is in line with the response to the previous question, that it is difficult to design such an architecture, but it is much easier to specify such capability through the form of a dataset. We have also read two past works suggested by the reviewer, in which they did analysis for verifying linguistic inductive biases. We have not yet come up with a good analogy for verifying reasoning inductive biases.
> > Nevertheless, we believe the reviewer has asked a very meaningful and important question. We will definitely keep this in mind and pursue the answer in future versions/works. Please also do not hesitate to suggest ways to validate this.
> >
> > “It’s not entirely clear to me that the specific tasks (Deduct, Induct, Abduct) will necessarily enforce the types of reasoning that they are intended to enforce.”
> >
> > We again thank the reviewer for posing a very deep question. We totally agree with the reviewer in that it is unclear if the specific tasks enforce the types of reasoning we want. We however note that deduction, induction, and abduction are high-level and philosophical concepts, and serve only as an inspiration for us to design the synthetic tasks. As we remarked at the beginning of section 3.2, “we design three synthetic tasks inspired by the three reasoning primitives ”. We do not expect the model will necessarily learn exactly these three capabilities. We also design tasks whose format mimics proving steps in theorem proving (see the rewrite task in Appendix B.1 for one example), which achieve very good results. That being said, we definitely appreciate the question and will keep it in mind, and look for ways to justify it more in future versions.
> >
> > "Other comments and citations"
> >
> > Thank you very much for your thorough review and providing many useful suggestions in terms of phrasing and citations. We have updated the paper accordingly.

---

> > > ### Comment · AnonReviewer3 · 2020-11-23
> > > **Thank you for the response!**
> > >
> > > Thank you for the detailed response! I appreciate the edits to the paper and believe that they strengthen the argument. Due to the (reasonable and understandable) limitations of the work done so far, I am keeping my score at 8 instead of raising it to 9 or 10, but I still think that the paper is very strong overall and provides several thought-provoking points that can inspire future work.

---

### Official Review · AnonReviewer4 · 2020-10-28
**Review for "LIME: LEARNING INDUCTIVE BIAS FOR PRIMITIVES OF MATHEMATICAL REASONING"**

**Rating:** 7
**Confidence:** 4

**Review:**


## Summary

The authors propose LIME: a pretraining strategy for learning inductive biases for mathematical reasoning. They construct 3 synthetic datasets corresponding to 3 basic reasoning patterns: deduction, induction, and abduction. Each dataset is designed to be devoid of concrete mathematical knowledge but encodes inductive biases of the reasoning pattern. In experiments, LIME pretraining improves the performance of a generic transformer model on 3 benchmarks for mathematical reasoning: IsarStep, HOList Skip-tree, and MetaMathStep.


## Strengths

+ I really like the idea of designing the synthetic datasets inspired by 3 basic patterns of mathematical reasoning: deduction, induction, and abduction. Though the basic reasoning patterns were discovered a while ago, it is novel to apply the idea to synthetic data for theorem proving. Traditional automated theorem proving focuses on deduction, but as the authors have explained in the paper, induction and abduction also play significant roles in conjecturing and framing new definitions.

+ The proposed LIME pretraining lead to descent performance improvements on 3 benchmarks. It is surprising since the pretraining task is a very simple string rewriting task without any mathematical knowledge. It is interesting if it actually works.

+ The paper is well-written and very easy to read.


## Weaknesses

- The authors acknowledged that there are numerous alternative designs for the pretraining datasets and presented a few alternatives in the appendix. Are there empirical comparisons between these alternatives? Why did the authors decide to use the current ones?

- It would be great to have ablations that only include 2 of the 3 pretraining datasets. Then we can know if the 3 reasoning patterns are truly irreducible.

- In Sec. 3.2, I'm confused about how the authors define "Rule" and "Case". To me, it looks like A * A + B = C is more like a "Case", whereas {A : a; B : b; C : d + e} is more like a "Rule"; since it describes how to map rule symbols to math symbols.

- I'm confused about framing "pretraining" as "learning inductive biases." Technically, I believe it's correct. Suppose "inductive biases" is defined as the learning algorithm's preferences when choosing one hypothesis among many equally valid ones. In that case, the initialization of model parameters is a kind of inductive bias.  Then, any pretraining is "learning inductive bias." It would be great if the authors clarify more about "pretraining" and "inductive biases" in the next revision.


## Conclusion

I like the idea of constructing synthetic datasets inspired by basic reasoning patterns. And it is surprising that pretraining on these simple tasks can work. However, I have a few questions, and the paper lacks ablations. I could potentially raise my score if the authors are able to address these questions.

## Minor Comments

These are minor points that did not account for my score. The authors do not have to address them in the rebuttal.

1. There should be more margin between Table 1 and the caption of Table 2.

2. Use "K" for thousand consistently, not "k."

3. "We set the size of the math symbol set 44 and rule symbol set of size 24." This sentence is difficult to parse correctly. I thought there was a typo before reading it a few more times.

---

> ### Author Response · Authors · 2020-11-20
> **Thank you for your inputs! We have run new experiments to answer your questions. We also provide further clarifications as follows.**
>
> We thank the reviewer for your efforts and suggestions. We are glad that the reviewer finds the paper interesting. The following are our answers to specific questions:
>
> “Are there empirical comparisons between these alternatives? Why did the authors decide to use the current ones”
>
> We include nine synthetic tasks in the updated version. In addition to the previous 4 synthetic tasks in the main paper, and several other variants described in the appendix, we further consider a multi-step version of the rewrite task, and a rewrite rule inference task, totalling up to 9 synthetic tasks. We provide a description of these extra results and the task description in appendix B. We used the current formulation because it is the simplest to define three tasks.
>
> “It would be great to have ablations that only include 2 of the 3 pretraining datasets. Then we can know if the 3 reasoning patterns are truly irreducible.”
>
> The irreducibility of the three tasks is a philosophical argument, and we would like to clarify that we do not believe that they are necessarily irreducible in practice. In fact, we found that there are cases in which training on one synthetic task can win overtraining on all three tasks. See Table 3, for example.
>
>
>
> “Case and Rule”
>
> A “Rule” expresses a certain law that one can apply in various circumstances. For example, consider the rule “if X is a male, then X is a human”. This rule can be applied to any male X, e.g., Jay, Peter, etc. X is just a placeholder for males’ names. Similarly, in our synthetic example, we let “Rule” to be an expression that has placeholders represented by rule symbols, and one can substitute any strings with. In your example, “A * A + B = C” is a rule in which one can substitute A, B, C with any random strings. Hence "A * A + B = C" is considered a Rule to us, and the substitution that corresponds to a particular circumstance, is considered as Case.
>
> Though we would like to point out that the exact definition of Rule and Case are not essential for this work, as we remarked there are various ways to define these synthetic tasks.
>
> “Clarification about inductive biases”
>
> By inductive biases, we mainly refer to the structural biases imposed by the architecture. We hypothesize that the self-attention layer in a transformer is a malleable structure that can represent all attention patterns, and learn various kinds of attention mechanisms. Such an attention pattern and mechanism can be thought of as a kind of architectural inductive bias. They encode the knowledge of how tokens in a sequence should attend to each other, in order to better process and integrate information. We believe such inductive bias or knowledge is general enough that it can be useful for a suite of tasks. We believe one can learn such inductive bias from one such task, and applies it to similar kinds of tasks. The evidence that LIME helps transformers much more than LSTMs supports our hypothesis. We hence believe by training on the synthetic tasks we defined, transformers learned useful architectural inductive biases for reasoning tasks.
>
> "Minor comments"
>
> Thanks for your suggestions! We have modified texts in our updated version.

---

> > ### Comment · AnonReviewer4 · 2020-11-21
> > **Updated Score**
> >
> > The authors address most of my questions. I would raise my score to 7. Though it would be more interesting if the 3 reasoning patterns were also irreducible in practice!

---

### Official Review · AnonReviewer2 · 2020-10-29
**Interesting perspective on designing pre-training datasets for automated theorem proving.**

**Rating:** 6
**Confidence:** 4

**Review:**

Summary:
This paper proposes a pretraining method, LIME, to improve Transformer’s performance on mathematical reasoning benchmarks. Specifically, this paper design three synthetic tasks to teach the transformer model to first learn three primitive reasoning steps: deduction, abduction, and induction respectively. The three datasets are used in the pre-training step, and the paper shows empirical performance gains on large mathematical reasoning tasks in the context of automated theorem proving.

Reasons for the score:
This paper provides an interesting direction in the field of automated theorem proving. In particular, it proposes a novel way to teach the model to learn primitive reasoning steps via designs of datasets. The paper provides good motivations and intuitions for their proposed method, but it would be nice for the authors to also formalize these intuitions and provide rigorous definitions as an academic paper (e.g. what is inductive bias and so on; details see below). The proposed method provides some ablation studies in the experiment section, but it is still worthwhile to conduct the following studies to enhance the quality of the paper: 1) how the positive transfer from the proposed pretraining change as we increase the size of the model? 2) It seems unclear from the current text how the LIME pretraining compares with large-scale unsupervised pretraining (e.g. BERT) in terms of convergence rate and the number of samples needed on the downstream tasks. For example, would pretraining on BERT lead to faster convergence and a smaller number of samples compared to LIME pretraining?

Pros:
1.The idea of designing pretraining datasets to help mathematical reasoning is novel.
2.Good empirical performance gains on large mathematical reasoning tasks in the context of automated theorem proving

Cons:
1.Though the proposed method provides some ablation studies in the experiment section, but it is still worthwhile to conduct the following studies to enhance the quality of the paper:

a. How the positive transfer from the proposed pretraining change as we increase the size of the model?

b. It seems unclear from the current text how the LIME pretraining compares with large-scale unsupervised pretraining (e.g. BERT) in terms of convergence rate and the number of samples needed on the downstream tasks. For example, would pretraining on BERT lead to faster convergence and a smaller sample complexity comparing to LIME pretraining?

c. Would the proposed method have similar improvements when using different architectures (e.g. LSTM, LSTM+attention, GPT2)?

2.The writing provides good intuitions in general, but it would be nice for the authors to also formalize these intuitions and provide rigorous definitions as an academic paper:

a. The word “inductive bias” may have different connotations across different contexts and it would be nice for the authors to give a more formal definition of what they mean by “inductive bias” in the context of this work. For example, inductive bias may refer to the structural bias imposed by the architecture. It may also refer to the prior knowledge of the target tasks or to the preference of the training scheme (e.g. what kind of functions are learned first by an SGD optimizer). It seems to me that the usage of inductive bias in this work is close to the prior knowledge of the target tasks, but this is not clear from the current writing.

b. Similarly, it would be nice for the authors to clearly state what they mean by ‘‘knowledge’’ in the context of this work. Is it referring to knowledge distillation, learned representations of the context, or something else? For example, in this sentence “we focus on the latter and design pre-training tasks that are intentionally devoid of knowledge and only allow the model to learn inductive bias for reasoning”, it is unclear why specific reasoning procedure cannot be treated as a form of knowledge.

3.The design of the three pre-training tasks is based on the three primitive rules for logical reasoning, which requires nontrivial human insights to formalize. Thus, the proposed approach may not be easily transferrable to other fields.

4.It would be nice for the authors to provide discussions with relevant works on the reasoning abilities of neural networks such as [1] and [2]. The proposed dataset in [1] could be of interest to this work as a pretraining dataset. [2] provides a theoretical framework to formalize the inductive bias in graph neural networks, which demonstrate the limitations of graph neural networks in terms of its reasoning capacity.

[1] Saxton, David, et al. "Analysing mathematical reasoning abilities of neural models." arXiv preprint arXiv:1904.01557 (2019).

[2] Xu, Keyulu, et al. "What Can Neural Networks Reason About?." arXiv preprint arXiv:1905.13211 (2019).

---

> ### Author Response · Authors · 2020-11-20
> **Thank you for your inputs! We have run new experiments to answer your questions. We also provide further clarifications as follows. Part I.**
>
> We thank the reviewer for your efforts and inputs. In the following are our answers to specific questions:
>
> “Does LIME help LSTMs, LSTM+attention, GPT2?”
>
> Firstly, we would like to point out that GPT2 is a pre-trained language model that uses transformer architecture (same as ours). For LSTMs, we have performed further experiments on studying the benefits of LIME. The new results are added to Appendix C.3 of the paper. We showed that LIME did improve LSTM and LSTM+attention, but the improvements were small compared to the transformer. Specifically, if we compare Top-1 accuracy, we can see that LIME improved LSTM from 5.5% to 6.9%, LSTM with attention from 12.3% to 13.4%, and transformer from 20.4% to 26.7%. This observation is aligned with our hypothesis that the transformer is a malleable architecture and hence it is capable of learning architectural inductive biases from datasets. This is mainly attributed to the potential of learning dynamic attention graphs in self-attention layers. We note that this still warrants further investigation as the performance of these architectures are not at the same level, and that may also lead to different improvements.
>
> “LIME vs BERT?”
>
> We find the question ambiguous with regards to the meaning of BERT. Does the reviewer ask for a BERT style method? Or an existing BERT pre-trained checkpoint? If the former, is the reviewer thinking of pre-training on natural language corpus, or formal math corpus? There are certain technical difficulties for each of these possibilities, and we summarize them as follows. But we find this is an interesting and useful research direction. We will include more results in future works.
>  - Firstly, BERT pretraining is often used for downstream tasks that do not need decoders, such as text classification tasks, because BERT only trains a transformer encoder. It is hence non-trivial to adapt a pre-trained BERT model for a sequence to sequence task like ours.
>  - Secondly, there are methods similar to BERT that do pre-training for the sequence to sequence tasks, such as MASS[1]. But we also cannot take one of the existing pre-trained checkpoints because our datasets require very different tokenization from the one used by the existing checkpoints. Note that LIME gets away with this issue by reinitializing the vocab embeddings for downstream tasks.
>  - Thirdly, BERT style pre-training usually is trained on a large natural language text corpus. It takes too long to finish training one pre-trained model from scratch before the rebuttal ends.
>  - Lastly, it may not be very interesting to pretrain the model on a common natural language dataset, as most of the texts are not math-related. But formal or informal math data is more scarce and hence more difficult to collect.
>
> “How does the positive transfer from the proposed pretraining change as we increase the size of the model?”
>
> We started some experiments to test the hypothesis that larger models also show positive transfer. We could not finish the experiment in time before the rebuttal ends. We will extend the paper with the results in future versions.
>
> [1] Song et. al., MASS: Masked Sequence to Sequence Pre-training for Language Generation.

---

> > ### Author Response · Authors · 2020-11-20
> > **Thank you for your inputs! We have run new experiments to answer your questions. We also provide further clarifications as follows. Part II.**
> >
> > “Clarification about inductive biases”
> >
> > By inductive biases, we mainly refer to the structural biases imposed by the architecture. We hypothesize that the self-attention layer in a transformer is a malleable structure that can represent all attention patterns, and learn various kinds of attention mechanisms. Such an attention pattern and mechanism can be thought of as a kind of architectural inductive bias. They encode the knowledge of how tokens in a sequence should attend to each other, in order to better process and integrate information. We believe such inductive bias or knowledge is general enough that it can be useful for a suite of tasks. We believe one can learn such inductive bias from one such task, and applies it to similar kinds of tasks. The evidence that LIME helps transformers much more than LSTMs supports our hypothesis. We hence believe by training on the synthetic tasks we defined, transformers learned useful architectural inductive biases for reasoning tasks.
> >
> > “Is it referring to knowledge distillation, learned representations of the context, or something else?”
> >
> > By knowledge, we mean domain-specific knowledge that is useful for the downstream tasks. For example, pre-training on Wikipedia may learn many world knowledge that helps with downstream question answering tasks. But LIME differs from this in that there is no domain-specific knowledge transfer from pre-training tasks to downstream tasks. We explicitly enforce this by reinitializing the vocabulary embeddings when loading pre-trained weights for downstream tasks, see section 5.2 for details. We agree with the reviewer that specific reasoning procedure is also a kind of knowledge, but it is different from the knowledge implied by the text: it is a kind of meta-knowledge or inductive bias that can be applied to a range of tasks.
> >
> > “The design of the three pre-training tasks is based on the three primitive rules for logical reasoning, which requires nontrivial human insights to formalize. Thus, the proposed approach may not be easily transferable to other fields.”
> >
> > We believe that logical/mathematical reasoning is a very general task. While LIME may not directly improve, say, image recognition or natural language processing, it is a general improvement of reasoning skills, and thereby it is clearly on a path that may have a huge impact across many different fields of science and technology.
> > In the short term, we believe that the conceptual contributions presented in this paper may directly affect other fields within machine learning: the design of synthetic, lightweight pretraining tasks to induce useful biases into transformers may turn out to be replicable in other fields.
> >
> > “It would be nice for the authors to provide discussions with relevant works on the reasoning abilities of neural networks such as [1] and [2].”
> >
> > Thanks for the pointers. We added a discussion of [1] and [2] to the related work section. The dataset presented in [1] requires to execute basic algorithms, such as addition, subtraction, multiplication, and division, and some symbolic (but easily computable) tasks such as solving small linear equations, partially evaluating functions, and computing the derivative of polynomials. This is quite different from the much more abstract mathematics in our other datasets that mostly consist of proofs. Another difference is the synthetic nature of the dataset in [1]. In principle, we could run LIME for their dataset, but we are not sure if there is much to learn from either a success or failure.
> >
> > [1] Saxton, David, et al. "Analysing mathematical reasoning abilities of neural models." arXiv preprint arXiv:1904.01557 (2019).
> >
> > [2] Xu, Keyulu, et al. "What Can Neural Networks Reason About?." arXiv preprint arXiv:1905.13211 (2019).

---

> > > ### Author Response · Authors · 2020-11-23
> > > **Does our rebuttal address your concern?**
> > >
> > > We wonder if our rebuttal have addressed your concerns. If not, we are happy to clarify further. Please let us know. Thank you very much.

---

### Author Response · Authors · 2020-11-20
**Thank all the reviewers for your efforts, great questions, and many editorial suggestions. We have updated the paper according to your suggestions.**

We would like to thank all the reviewers for your efforts, great questions, and many editorial suggestions. We are encouraged by many of the positive reviews. We also updated our paper according to your suggestion. The following is a list of updates we have made to the newest version.

- *Appendix B*:
    - Added description of two new tasks: rewrite_multistep (B.1), induct_rewrite (B.2).
    - Added the full table results on all 9 pre-training tasks (B.3).
- *Appendix C*:
    - Added the results on applying LIME to LSTMs (C.3)
- *Main paper*:
    - Introduction: edited the arguments related to “strong human bias” (suggested by R3). Added citations suggested by R3.
    - Related work section: Added a paragraph on discussing few related works on inductive biases, recommended by R2 and R3.
    - Many minor edits including adding citations, fixing typos, fixing reference, changing phrases suggested by R2, R3 and R4.

---

### Decision · Program_Chairs · 2021-01-07
**Final Decision**

**Decision:**

Reject

**Comment:**


The authors propose a pretraining strategy learning inductive biases in transformers for deduction, induction, and abduction.  Further, the claims and results seem to indicate that such pretraining is more successful in transformers which provide a more malleable architecture for learning inductive (structural) biases.  There are open questions that remain, specifically surrounding disentangling high performance from structural bias learning (i.e. is pretraining doing what we think it is) and whether datasets are the "correct" mechanism for imparting such biases/knowledge.